# FaDIn: Fast Discretized Inference for Hawkes Processes with General Parametric Kernels

## Abstract

Temporal point processes (TPP) are a natural tool for modeling event-based data. Among all TPP models, Hawkes processes have proven to be the most widely used, mainly due to their adequate modeling for various applications, in particular when considering exponential or non-parametric kernels. Although non-parametric kernels are an option, such models require large datasets. While exponential kernels are more data efficient and relevant for certain applications where events immediately trigger more events, they are ill-suited for applications where latencies need to be estimated, such as in neuroscience. This work aims to offer an efficient solution to TPP inference using general parametric kernels with finite support. The developed solution consists of a fast L2 gradient-based solver leveraging a discretized version of the events. After supporting the use of discretization theoretically, the statistical and computational efficiency of the novel approach is demonstrated through various numerical experiments. Finally, the effectiveness of the method is evaluated by modeling the occurrence of stimuli-induced patterns from brain signals recorded with magnetoencephalography (MEG). Given the use of general parametric kernels, results show that the proposed approach leads to a more plausible estimation of pattern latency compared to the state-of-the-art.

## 1 Introduction

The statistical framework of Temporal Point Processes (TPPs; see *e.g.,* Daley & Vere-Jones 2003) is well adapted for modeling event-based data. It offers a principled way to predict the rate of events as a function of time and the previous events' history. TPPs are historically used to model intervals between events, such as in renewal theory, which studies the sequence of intervals between successive replacements of a component susceptible to failure. TPPs find many applications in neuroscience, in particular, to model single-cell recordings and neural spike trains (Truccolo et al., 2005; Okatan et al., 2005; Kim et al., 2011; Rad & Paninski, 2011), occasionally associated with spatial statistics (Pillow et al., 2008) or network models (Galves & Löcherbach, 2015). In the machine learning community, there is a growing interest in these statistical tools (Bompaire, 2019; Shchur et al., 2020; Mei et al., 2020). Multivariate Hawkes processes (MHP; Hawkes 1971) are likely the most popular, as they can model interactions between each univariate process. They also have the peculiarity that a process can be self-exciting, meaning that a past event will increase the probability of having another event in the future on the same process. The conditional intensity function is the key quantity for TPPs. With MHP, it is composed of a baseline parameter and kernels. It describes the probability of occurrence of an event depending on time. The kernel function represents how processes influence each other or themselves. The most commonly used inference method to obtain the baseline and the kernel parameters of MHP is the maximum likelihood (MLE; see *e.g.,* Daley & Vere-Jones, 2007 or Lewis & Mohler, 2011). One alternative and often overlooked estimation criterion is the least squares $\ell_2$ error, inspired by the theory of empirical risk minimization (Reynaud-Bouret & Rivoirard, 2010; Hansen et al., 2015; Bacry et al., 2020).

A key feature of MHP modeling is the choice of kernels. Non-parametric and parametric kernels are the two possibilities. In the non-parametric setting, kernel functions are approximated by histograms (Lewis & Mohler, 2011; Lemonnier & Vayatis, 2014), by a linear combination of pre-defined functions (Zhou et al., 2013a; Xu et al., 2016), by functions lying in a RKHS (Yang et al., 2017) or, alternatively, by neural networks (Mei & Eisner, 2017; Shchur et al., 2019; Pan et al., 2021). In addition to the frequentist approach, many Bayesian approaches, such as Gibbs sampling (Ishwaran & James,

2001) or (stochastic) variational inference (Hoffman et al., 2013), have been adapted to MHP in particular to fit non-parametric kernels. Bayesian methods also rely on the modelling of the kernel by histograms (*e.g.,* Donnet et al., 2020) or by a linear combination of pre-defined functions (*e.g.,* Linderman & Adams, 2015). These approaches are designed whether in continuous-time (Rasmussen, 2013; Zhang et al., 2018; Donnet et al., 2020; Sulem et al., 2021) or in discrete-time (Mohler et al., 2013; Linderman & Adams, 2015; Zhang et al., 2018; Browning et al., 2022). These functions allow great flexibility for the shape of the kernel, yet this comes at the risk of poor estimation of it when only a small amount of data is available (Xu et al., 2017). Another approach to estimate the intensity function is to consider kernels parametrized by $\eta$. Although it can introduce a potential bias by assuming a particular shape for kernels, this approach has several benefits. First, it reduces inference difficulties , as $\eta$ is typically lower dimensional compared to non-parametric kernels. Moreover, for kernels satisfying the Markov property (Bacry et al., 2015), computing the conditional intensity function is linear in the total number of timestamps/events. The most popular kernel belonging to this family is the exponential kernel (Ogata, 1981). It is defined by $\eta = (\alpha, \gamma) \mapsto \alpha\gamma \exp(-\gamma t)$, where $\alpha$ and $\gamma$ are the scaling and the decay parameters, respectively (Veen & Schoenberg, 2008; Zhou et al., 2013b). However, as pointed out by Lemonnier & Vayatis (2014), the maximum likelihood estimator for MHP with exponential kernels is efficient only if the decay $\gamma$ is fixed. Thus, only the scaling parameter is usually inferred. This implies that the hyperparameter $\gamma$ must be chosen in advance, usually by using a grid search, a random search, or Bayesian optimization. This leads to a computational burden when the dimension of the MHP is high. The second option is to define a $\gamma$ decay parameter common to all kernels, which results in a loss of expressiveness of the model. In both cases, the relevance of the exponential kernel relies on the choice of the decay parameter, which may not be adapted to the data (Hall & Willett, 2016). For more general parametric kernels which do not verify the Markov property, the inference procedure with both MLE or $\ell_2$ loss scales poorly as they have quadratic computational scaling with the number of events, making their use limited in practice (see *e.g.,* Bompaire, 2019, Chapter 1). These limitations for parametric and non-parametric kernels prevent their usage in some applications, as pointed out by Carreira (2021) in finance or Allain et al. (2021) in neuroscience. A strong motivation for this work is also neuroscience applications.

The quantitative analysis of electrophysiological signals such as electroencephalography (EEG) or magnetoencephalography (MEG) is a challenging modern neuroscience research topic (Cohen, 2014). By giving a non-invasive way to record human neural activity with a high temporal resolution, EEG and MEG offer a unique opportunity to study cognitive processes as triggered by controlled stimulation (Baillet, 2017). Convolutional dictionary learning (CDL) is an unsupervised algorithm that has recently been proposed to study M/EEG signals (Jas et al., 2017; Dupré la Tour et al., 2018). It consists in extracting patterns of interest in M/EEG signals. It learns a combination of time-invariant patterns – called *atoms* – and their activation function to reconstruct the signal sparsely. However, while CDL recovers the local structure of signals, it does not provide any global information, such as interactions between patterns or how their activations are affected by stimuli. Atoms typically correspond to transient bursts of neural activity (Sherman et al., 2016) or artifacts such as eye blinks or heartbeats. By offering an event-based perspective on non-invasive electromagnetic brain signals, CDL makes Hawkes processes amenable to M/EEG-based studies. Given the estimated events, one important goal is then to uncover potential temporal dependencies between external stimuli presented to the subject and the appearance of the atoms in the data. More precisely, one is interested in statistically quantifying such dependencies, *e.g.,* by estimating the mean and variance of the neural response latency following a stimulus. In Allain et al. (2021), the authors address this precise problem. Their approach is based on an EM algorithm and a Truncated Gaussian kernel, which can cope with only a few brain data, as opposed to non-parametric kernels, which are more data hungry. Beyond neuroscience, Carreira (2021) use a likelihood-based approach using exponential kernels to model order book events. Their approach use high-frequency trading data, taking account of latency at hand in the proposed loss.

This paper proposes a new inference method – named FaDIn – to estimate any parametric kernels for Hawkes processes. Our approach is based on two key features. First, we use finite-support kernels and a discretization applied to the ERM-inspired least-squares loss. Second, we propose to employ some precomputations that significantly reduce the computational cost. We then show that the implicit bias induced by the discretization procedure is negligible compared to the statistical error. Further, we highlight the efficiency of FaDIn in computation and statistical estimation over the non-parametric approach. Finally, we demonstrate the benefit of using a general kernel with MEG

data. The flexibility of FaDIn allows us to model neural response to external stimuli with a much better-adapted kernel than the existing method derived in Allain et al. (2021).

## 2 FAST DISCRETIZED INFERENCE FOR HAWKES PROCESSES (FADIN)

### 2.1 HAWKES PROCESSES

Given a stopping time $T \in \mathbb{R}_+$ and an observation period $[0, T]$, a temporal point process (TPP) is a stochastic process whose realization consists of a set of distinct timestamps $\mathscr{F}_T = \{t_n, \ t_n \in [0, T]\}$ occurring in continuous time. The behavior of a TPP is fully characterized by its *intensity function* that corresponds to the expected infinitesimal rate at which events are occurring at time $t \in [0, T]$. The values of this function may depend on time (*e.g.,* inhomogeneous Poisson processes) or rely on past events such as self-exciting processes (see Daley & Vere-Jones 2003 for an excellent account of TPP). For the latter, the occurrence of one event will modify the probability of having a new event in the near future. The conditional intensity function $\lambda : [0, T] \to \mathbb{R}_+$ have the following form:

$$\lambda \left( t | \mathscr{F}_t \right) := \lim_{\mathrm{d}t \to 0} \frac{\mathbb{P} \left( N_{t+\mathrm{d}t} - N_t = 1 | \mathscr{F}_t \right)}{\mathrm{d}t} \ ,$$

where $N_t := \sum_{n \geq 1} \mathbf{1}_{t_n \leq t}$ is the counting process associated to the PP. Among this family, Multivariate Hawkes processes (MHP; Hawkes, 1971) model the interactions of $p \in \mathbb{N}_*$ self-exciting TPPs. Given $p$ sets of timestamps $\mathscr{F}_T^i = \{t_n^i, \ t_n^i \in [0, T]\}_{n=1}^{N_T^i}, i = 1, \ldots, p$ , each process is described by the following intensity function:

$$\lambda_i(t) = \mu_i + \sum_{j=1}^{p} \int_0^t \phi_{ij}(t - s) \, \mathrm{d}N_s^j \ , \tag{1}$$

where $\mu_i$ is the baseline parameter, $N_t = [N_t^1, \ldots, N_t^p]$ the associated multivariate counting process and $\phi_{ij} : [0, T] \to \mathbb{R}_+$ the excitation function – called *kernel* – representing the influence of $j$-th process' past events onto $i$-th process' future events. From an inference perspective, the goal is to estimate the baseline and kernels associated with the MHP from the data. In this paper, we focus on the ERM-inspired least squares loss. Assuming a class of parametric kernel parametrized by $\eta$, the objective is to find parameters that minimize (see *e.g.,* Eq. (I.2) in Bompaire, 2019, Chapter 1):

$$\mathcal{L} \left( \theta, \mathscr{F}_T \right) = \frac{1}{N_T} \sum_{i=1}^{p} \left( \int_0^T \lambda_i(s)^2 \, \mathrm{d}s - 2 \sum_{t_n^i \in \mathscr{F}_T^i} \lambda_i \left( t_n^i \right) \right) , \tag{2}$$

where $N_T = \sum_{i=1}^{p} N_T^i$ is the total number of timestamps, and where we denote by $\theta = (\mu, \eta)$. Interestingly, when used with an exponential kernel, this loss benefits from some precomputations of complexity $O(N_T)$, making the subsequent iterative optimization procedure independent of $N_T$. This computational ease is the main advantage of the loss $\mathcal{L}$ over the log-likelihood function. However, when using a general parametric kernel, these precomputations require $O((N_T)^2)$ operations killing the computational benefit of the $\ell_2$ loss $\mathcal{L}$ over the log-likelihood. It is worth noting that this loss differs from the quadratic error minimized between the counting processes and the integral of the intensity function, as used in Wang et al. (2016); Eichler et al. (2017) and Xu et al. (2018).

### 2.2 FADIN

The approach we propose in this paper fills the need for general parametric kernels in many applications. We provide a computationally and statistically efficient solver – coined FaDIn – that works with many parametric kernels using gradient-based algorithms. Precisely, it relies on the three key ideas: (*i*) the use of parametric finite-support kernels, (*ii*) a discretization of the time interval $[0, T]$, and (*iii*) precomputations allowing an efficient optimization procedure detailed below.

**Finite support kernels** A core bottleneck for MLE or $\ell_2$ estimation of parametric kernels is the need to compute the intensity function for all events. For general kernels, the intensity function usually requires $O((N_T)^2)$ operations, which makes it intractable for long-time length processes. To make this computation more efficient, we consider finite support kernels. Using a finite support

kernel amounts to setting a limit in time for the influence of a past event on the intensity, *i.e.,* $\forall t \notin [0, W], \phi_{ij}(t) = 0$, where $W$ denotes the length of the kernel's support. This assumption matches applications where an event cannot have influence far in the future, such as in neuroscience (Krumin et al., 2010; Eichler et al., 2017; Allain et al., 2021) or high-frequency trading (Bacry et al., 2015; Carreira, 2021). The intensity function (3) can then be reformulated as a convolution between the kernel $\phi_{ij}$ and the sum of Dirac functions $z_i := \sum_{t_n^i \in \mathscr{F}_T^i} \delta_{t_n^i}$ located at the event occurrences $t_n^i$:

$$\lambda_i(t) = \mu_i + \sum_{j=1}^{p} \phi_{ij} * z_j(t), \quad t \in [0, T] \quad .$$

As $\phi_{ij}$ has a finite support, the intensity can be computed efficiently with this formula. Indeed, only events in the interval $[t - W, t]$ need to be considered. This usually amounts to a fraction of the events of the full process.

**Discretization** To make these computations even more efficient, we propose to rely on discretized processes. Most Hawkes processes estimation procedures involve a continuous paradigm to minimize (2) or its log-likelihood counterpart. Discretization has been investigated so far for non-parametric kernels (Kirchner, 2016; Kirchner & Bercher, 2018; Kurisu, 2018). The discretization of a TPP consists in projecting each event $t_n^i$ on a regular grid $\mathcal{G} = \{0, \Delta, 2\Delta, \dots, G\Delta\}$, where $G = \lfloor \frac{T}{\Delta} \rfloor$. We refer to $\Delta$ as the stepsize of the discretization. Here $\lfloor \cdot \rfloor$ denotes the floor function. Let $\widetilde{\mathscr{F}}_T^i$ be the set of projected timestamps of $\mathscr{F}_T^i$ on the grid $\mathcal{G}$. The intensity function of the $i$-th process of our discretized MHP is defined as:

$$\tilde{\lambda}_i[s] = \mu_i + \sum_{j=1}^{p} \sum_{\tilde{t}_m^j \in \widetilde{\mathscr{F}}_{s\Delta}^j} \phi_{ij}(s\Delta - \tilde{t}_m^j) = \mu_i + \sum_{j=1}^{p} \underbrace{\sum_{\tau=1}^{L} \phi_{ij}^{\Delta}[\tau] z_j[s - \tau]}_{(\phi_{ij}^{\Delta} * z_j)[s]}, \quad s \in [\![0, G]\!] \quad , \qquad (3)$$

where $L = \lfloor \frac{W}{\Delta} \rfloor$ denotes the number of points on the discretized support, $\phi_{ij}^{\Delta}[s] = \phi_{ij}(s\Delta)$ is the kernel value on the grid and $z_i[s] = \# \left\{ |t_n^i - s\Delta| \leq \frac{\Delta}{2} \right\}$ denotes the number of events projected on the grid timestamp $s$. From now and throughout the rest of the paper, we denote $\phi_{ij}(\cdot) : \mathbb{R}_+ \to \mathbb{R}_+$ as a function while $\phi_{ij}^{\Delta}[\cdot]$ represents the discrete vector $\phi_{ij}^{\Delta} \in \mathbb{R}_+^L$. Compared to the continuous formulation, the intensity function can be computed more efficiently as one can rely on discrete convolutions, whose worst case complexity scales as $O(N_T L)$. It can also be further accelerated using Fast Fourier Transform when $N_T$ is large. Another benefit of the discretization is that for kernel whose values are costly to compute, at most $L$ values need to be computed. This can have a strong computational impact when $N_T \gg L$ as all values can be precomputed and stored.

While discretization improves the computational efficiency, it also introduces a bias in the computation of the intensity function and, thus potentially, in estimating the kernel parameters. The impact of the discretization on the estimation is considered in Section 2.3 and Section 3.1. Note that this bias is similar to the one incurred by quantizing the kernel as histograms for non-parametric estimators.

**Loss and precomputations** FaDIn aims at minimizing the discretized $\ell_2$ loss, which approximates the integral on the left part of (2) by a sum on the grid $\mathcal{G}$ after projecting timestamps of $\mathscr{F}_T$ on it. It boils down to optimizing the following loss:

$$\mathcal{L}_{\mathcal{G}}\left(\theta, \widetilde{\mathscr{F}}_T\right) = \frac{1}{N_T} \sum_{i=1}^{p} \left( \Delta \sum_{s \in [\![0, G]\!]} \left(\tilde{\lambda}_i[s]\right)^2 - 2 \sum_{\tilde{t}_n^i \in \widetilde{\mathscr{F}}_T^i} \tilde{\lambda}_i \left[ \frac{\tilde{t}_n^i}{\Delta} \right] \right) \quad . \qquad (4)$$

To find the parameters of the intensity function $\theta$, FaDIn minimizes $\mathcal{L}_{\mathcal{G}}$ using a first-order gradient-based algorithm. The computational bottleneck of the proposed algorithm is thus the computation of the gradient $\nabla \mathcal{L}_{\mathcal{G}}$ regarding parameters $\theta$. Using the discretized finite-support kernel, this gradient can be computed using convolution, giving the same computational complexity as the computation of the intensity function $O(N_T L)$.

However, gradient computation can still be too expensive for long processes with many events to get reasonable inference times. Using the least squares error of the process (4), one can further reduce

the complexity of computing the gradient by precomputing some constants $\Phi_j(\tau; G)$, $\Psi_{j,k}(\tau, \tau'; G)$ and $\Phi_j(\tau; \widetilde{\mathscr{F}}_T^i)$ that do not depend on the parameter $\theta$. Indeed, by developing and rearranging the terms in (4), one obtains:

$$
N_T \, \mathcal{L}_\mathcal{G}\left(\theta, \widetilde{\mathscr{F}}_T\right) = T\|\mu\|_2^2 + 2\Delta \sum_{i=1}^{p} \mu_i \sum_{j=1}^{p} \sum_{\tau=1}^{L} \phi_{ij}^\Delta[\tau] \underbrace{\left(\sum_{s=1}^{G} z_j[s-\tau]\right)}_{\Phi_j(\tau; G)}
$$

$$
+ \Delta \sum_{i=1}^{p} \sum_{k=1}^{p} \sum_{j=1}^{p} \sum_{\tau=1}^{L} \sum_{\tau'=1}^{L} \phi_{ij}^\Delta[\tau] \phi_{ik}^\Delta[\tau'] \underbrace{\left(\sum_{s=1}^{G} z_j[s-\tau]\, z_k[s-\tau']\right)}_{\Psi_{j,k}(\tau,\tau'; G)}
$$

$$
- 2\left(\sum_{i=1}^{p} N_T^i \mu_i + \sum_{i=1}^{p} \sum_{j=1}^{p} \sum_{\tau=1}^{L} \phi_{ij}^\Delta[\tau] \underbrace{\left(\sum_{\tilde{t}_n^i \in \tilde{\mathscr{F}}_T^i} z_j\left[\frac{\tilde{t}_n^i}{\Delta} - \tau\right]\right)}_{\Phi_j\left(\tau; \widetilde{\mathscr{F}}_T^i\right)}\right) \ .
$$

The term $\Psi_{j,k}(\tau, \tau'; G)$ dominates the computational cost of our precomputations. It requires $O(G)$ operations for each tuples $(\tau, \tau')$ and $(j, k)$. Thus, it has a total complexity of $O(p^2 L^2 G)$ and is the bottleneck of the precomputation phase. For any $m \in \{1, \ldots, p\}$, the gradient of the loss w.r.t. the baseline parameter is given by:

$$
N_T \frac{\partial \mathcal{L}_G}{\partial \mu_m} = 2T\mu_m - 2N_T^m + 2\Delta \sum_{j=1}^{p} \sum_{\tau=1}^{L} \phi_{mj}^\Delta[\tau] \Phi_j(\tau; G) \ .
$$

For any tuple $(m, l) \in \{1, \ldots, p\}^2$, the gradient of $\eta_{ml}$ is:

$$
N_T \frac{\partial \mathcal{L}_G}{\partial \eta_{ml}} = 2\Delta \mu_m \sum_{\tau=1}^{L} \frac{\partial \phi_{m,l}^\Delta[\tau]}{\partial \eta_{m,l}} \Phi_l(\tau; G) + 2\Delta \sum_{k=1}^{p} \sum_{\tau=1}^{L} \sum_{\tau'=1}^{L} \phi_{mk}^\Delta[\tau'] \frac{\partial \phi_{m,l}^\Delta[\tau]}{\partial \eta_{m,l}} \Psi_{l,k}(\tau, \tau'; G)
$$

$$
- 2 \sum_{\tau=1}^{L} \frac{\partial \phi_{m,l}^\Delta[\tau]}{\partial \eta_{m,l}} \Phi_l(\tau; \widetilde{\mathscr{F}}_T^m) \ .
$$

Gradients of kernel parameters dominate the computational cost of gradients. The complexity is of $O(pL^2)$ for each kernel parameter, leading to a total complexity of $O(p^3 L^2)$ and is independent of the number of events $N_T$. Thus, a trade-off can be made between the precision of the method and its computational efficiency when varying the size of the kernel's support or the discretization.

**Optimization** The inference is then conducted using gradient descent for the $\ell_2$ loss $\mathcal{L}_G$. FaDIn thus allows for very general parametric kernels, as exact gradients for each parameter involved in the kernels can be derived efficiently as long as the kernel is differentiable and has a finite support. Gradient-based optimization algorithms can therefor be used without limitation, in contrast with the EM algorithm which requires close form solution to zero the gradient, which is difficult for many kernels. An important remark is that the problem is generally non-convex and may converge to a local minimum.

### 2.3 IMPACT OF THE DISCRETIZATION

While discretization allows for efficient computations, it also introduces a perturbation in the loss value. In this section, we quantify the impact of this perturbation on the parameter estimation when $\Delta$ goes to 0. Through this section, consider we observe a process $\mathscr{F}_T$ whose intensity function is given by the parametric form $\lambda(\cdot; \theta^*)$. Note that if the process $\mathscr{F}_T$'s intensity is not in the parametric family $\lambda(\cdot; \theta)$, $\theta^*$ is defined as the best approximation of its intensity function in the $\ell_2$ sense. The goal of the inference process is thus to recover the parameters $\theta^*$.

When working with the discrete process $\widetilde{\mathscr{F}}_T$, the events $t_n^i$ of the original process are replaced with a projection on a grid $\tilde{t}_n^i = t_n^i + \delta_n^i$. Here, $\delta_n^i$ is uniformly distributed on $[-\Delta/2, \Delta/2]$. We consider

the discrete FaDIn estimator $\widehat{\theta}_\Delta$ defined as $\widehat{\theta}_\Delta = \arg\min_\theta \mathcal{L}_\mathcal{G}(\theta)$. We can upper-bound the error incurred by $\widehat{\theta}_\Delta$ by the decomposition:

$$\left\|\widehat{\theta}_\Delta - \theta^*\right\|_2 \leq \underbrace{\left\|\widehat{\theta}_c - \theta^*\right\|_2}_{(*)} + \underbrace{\left\|\widehat{\theta}_\Delta - \widehat{\theta}_c\right\|_2}_{(**)} , \qquad (5)$$

where $\widehat{\theta}_c = \arg\min_\theta \mathcal{L}(\theta)$ is the reference estimator for $\theta^*$ based on the standard $\ell_2$ estimator for continuous point processes. This decomposition involves the statistical error $(*)$ and the bias error $(**)$ induced by the discretization. The statistical term measures how far the parameters obtained by minimizing the $\ell_2$ continuous loss having access to a finite amount of data are from the true ones. In contrast, the term $(**)$ represents the discretization bias induced by minimizing the discrete loss (4) instead of the continuous one (2). In the following proposition, we focus on the discretization error $(**)$ which is related to the computational trade-off offered by our method and not on the statistical error of the continuous $\ell_2$ estimator $(**)$. Our work showcases that this disregarded estimator can be efficiently computed, and we hope it will promote research to describe its asymptotic behavior. We now study the perturbation of the loss due to the discretization.

**Proposition 1.** *Let $\mathscr{F}_T$ and $\widetilde{\mathscr{F}_T}$ be respectively a MHP process and its discretized version on a grid $\mathcal{G}$ with stepsize $\Delta$. Assume that the intensity function of $\mathscr{F}_T$ possesses continuously differentiable finite support kernels on $[0, W]$. Thus, assuming $\Delta < \min_{t_n^i, t_m^j \in \mathscr{F}_T} |t_n^i - t_m^j|$, for any $i \in [\![1, p]\!]$, it holds:*

$$\widetilde{\lambda}_i[s] = \lambda_i(s\Delta) - \sum_{j=1}^p \sum_{t_m^j \in \mathscr{F}_{s\Delta}^j} \delta_m^j \frac{\partial \phi_{ij}}{\partial t}(s\Delta - t_m^j; \theta) + O(\Delta^2) ,$$

$$\mathcal{L}_\mathcal{G}(\theta) = \mathcal{L}(\theta) + \Delta.h(\theta) + \frac{2}{N_T} \sum_{i=1}^p \sum_{t_n^i \in \mathscr{F}_T^i} \sum_{j=1}^p \sum_{t_m^j \in \mathscr{F}_{s\Delta}^j} (\delta_m^j - \delta_n^i) \frac{\partial \phi_{ij}}{\partial t}(t_n^i - t_m^j; \theta) + O(\Delta^2) .$$

The technical proof is deferred to Section B.1 in the Appendix. The first result is a direct application of the Taylor expansion of the intensity for the kernels. For the loss, the first perturbation term $\Delta.h(\theta)$ comes from approximating the integral with a finite Euler sum (Tasaki, 2009) while the second one derives from the perturbation of the intensity. This proposition shows that as the discretization step $\Delta$ goes to 0, the perturbed intensity and $\ell_2$ loss are good estimates of their continuous counterpart. We now quantify the discretization error $(**)$ as $\Delta$ goes to 0.

**Proposition 2.** *We consider the same assumption as in Proposition 1. Then, if the estimators $\widehat{\theta}_c = \arg\min_\theta \mathcal{L}(\theta)$ and $\widehat{\theta}_\Delta = \arg\min_\theta \mathcal{L}_\mathcal{G}(\theta)$ are uniquely defined, $\widehat{\theta}_\Delta$ converges to $\widehat{\theta}_c$ as $\Delta \to 0$. Moreover, if $\mathcal{L}$ is $C^2$ and its hessian $\nabla^2 \mathcal{L}(\widehat{\theta}_c)$ is positive definite with $\varepsilon > 0$ its smallest eigenvalue, then $\|\widehat{\theta}_\Delta - \widehat{\theta}_c\|_2 \leq \frac{\Delta}{\varepsilon} g(\widehat{\theta}_\Delta)$, with $g(\widehat{\theta}_\Delta) = O(1)$.*

This proposition shows that asymptotically on $\Delta$, the estimator $\widehat{\theta}_\Delta$ is equivalent to $\widehat{\theta}_c$. It also shows that the discrete estimator converges to the continuous one at the same speed as $\Delta$ decreases. This is confirmed experimentally by results shown in Figure A.7 in the Appendix. Thus, one would need to select $\Delta$ so that the discretization error is small compared to the statistical one.

## 3 NUMERICAL EXPERIMENTS

We present various synthetic data experiments to support the advantages of the proposed approach. To begin, we investigate the bias induced by the discretization in Section 3.1. Afterwards, the statistical and computational efficiency of FaDIn is highlighted through a benchmark with popular non-parametric approaches Section 3.2. Due to the space limitation, sensitivity analysis regarding the parameter $W$ and additional non-parametric comparisons are provided in Section A.1 and Section A.2, respectively.

### 3.1 CONSISTENCY OF DISCRETIZATION

In order to study the estimation bias due to discretization, we run two experiments and report the results in Figure 1 (details and further experiments are presented in Section A.3 and Section A.4

in the Appendix). The general paradigm is a one-dimensional TPP with intensity parametrized as in (1) with a Truncated Gaussian kernel of mean $m \in \mathbb{R}$ and standard deviation $\sigma > 0$, with fixed support $[0, W] \subset \mathbb{R}^+$, $W > 0$. It corresponds to $\phi(\cdot) = \alpha\kappa(\cdot), \alpha \geq 0$ with

$$\kappa(\cdot) := \kappa(\cdot; m, \sigma, W) = \frac{1}{\sigma} \frac{f\left(\frac{\cdot - m}{\sigma}\right)}{F\left(\frac{W-m}{\sigma}\right) - F\left(\frac{-m}{\sigma}\right)} \mathbf{1}_{0 \leq \cdot \leq W} \ ,$$

where $f$ (resp. $F$) is the probability density function (resp. cumulative distribution function) of the standard normal distribution. Hence, the parameters to estimate are $\mu$ and $\eta = (\alpha, m, \sigma)$.

In both experiments, for multiple process length $T$, the discrete estimates are computed for varying grid stepsize $\Delta$, from $10^{-1}$ to $10^{-3}$. The parameter $W$ is set to 1. The $\ell_2$ norm of the difference between estimates and the true parameter values –the ones used for data simulation– is computed and reported. We first computed the parameter estimates with our FaDIn method for $T \in \{10^3, 10^5, 10^4, 10^6\}$, for 100 simulations each time. Second, since we wish to separate discretization bias from statistical bias, we compute the estimates with an EM algorithm, both continuously and discretely, and that for 50 random data simulations. For the latter, the process is not self-excited, but rather driven by an exogenous homogeneous Poisson Process (Allain et al., 2021).

One can observe that the $\ell_2$ errors between discrete estimates and true parameters tend towards zero as $T$ increases. For $T$ fixed, one can see plateaus starting for stepsize values that are not particularly small, indicating that the discretization bias is limited. The second experiment with the EM algorithm shows that when the plateau mentioned above is reached, it corresponds to some statistical error. In other words, even for a reasonably coarse stepsize, the bias induced by the discretization is slight compared to the statistical error.

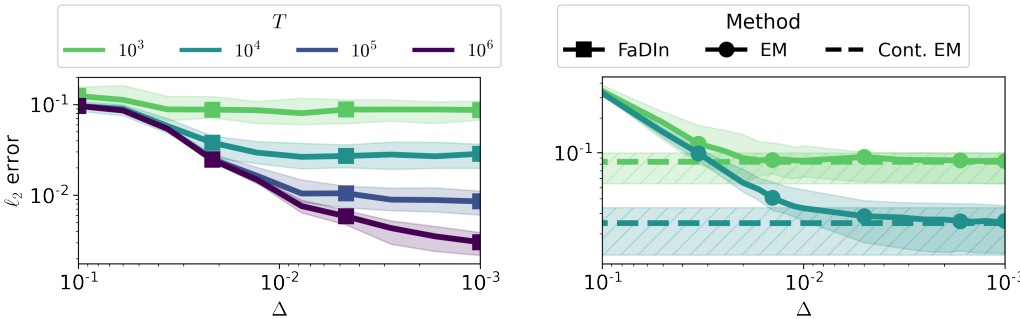

Figure 1: Median and interquartile error bar of the $\ell_2$ norm between true parameters and parameter estimates computed with FaDIn (left) and with EM algorithm (right), continuously and discretely, w.r.t. the stepsize of the grid $\Delta$.

## 3.2 STATISTICAL AND COMPUTATIONAL EFFICIENCY OF FaDIn

We compare FaDIn with non-parametric methods by assessing approaches' statistical and computational efficiency. To learn the non-parametric kernel, we select various existing methods. The first benchmarked method uses histogram kernels and relies on the EM algorithm, provided in Zhou et al. (2013a) and implemented in the `tick` library (Bacry et al., 2017). The kernel is set with one basis function. The three other approaches involve a linear combination of pre-defined raised cosine functions as non-parametric kernels. The inference is made either by stochastic gradient descent algorithm (Non-param SGD; Linderman & Adams, 2014) or by Bayesian approaches such as Gibbs sampling (Gibbs) or Variational Inference (VB) from Linderman & Adams (2015). These algorithms are implemented in the `pyhawkes` library. In the following experiments, we set the number of basis to five for each method. More precisely, we simulate a two-dimensional Hawkes process (repeated ten times) using the `tick` library with baseline $\boldsymbol{\mu} = [0.1, 0.2]$ and Raised Cosine kernels:

$$\phi_{i,j}(\cdot) = \alpha_{i,j}\left[1 + \cos\left(\frac{\cdot - u_{i,j}}{\sigma_{i,j}}\pi - \pi\right)\right]\mathbb{I}\left\{\cdot \in [u_{i,j}; u_{i,j} + 2\sigma_{i,j}]\right\} \ , (i,j) \in \{1,2\}^2$$

with parameters $\boldsymbol{\alpha} = \begin{bmatrix} 1.5 & 0.1 \\ 0.1 & 1.5 \end{bmatrix}$, $\mathbf{u} = \begin{bmatrix} 0.1 & 0.3 \\ 0.3 & 0.3 \end{bmatrix}$ and $\boldsymbol{\sigma} = \begin{bmatrix} 0.3 & 0.25 \\ 0.3 & 0.3 \end{bmatrix}$. Further, we infer the intensity function of these underlying Hawkes processes using FaDIn and the four previously

mentioned methods setting $\Delta = 0.01$ for all these discrete approaches. This experiment is repeated for varying values of $T \in \{10^3, 10^4, 10^5\}$. The averaged (over the ten runs) normalized $\ell_1$ error on the intensity (evaluated on the same discrete grid), as well as the associated computation time, are reported in Figure 2. From a statistical perspective, we can observe the advantages of FaDIn inference for varying $T$ over the benchmarked methods. It is worth noting that this result is expected by a parametric approach when the used kernel belongs to the same family as the one with which events have been simulated. From a computational perspective, FaDIn is very efficient compared to benchmarked approaches. Indeed, it scales very well with an increasing time $T$ and then with a growing number of events. In contrast, other methods depend on the number of events and scale linearly with the time $T$. For completeness, different kernel shapes are provided in Section A.2.1.

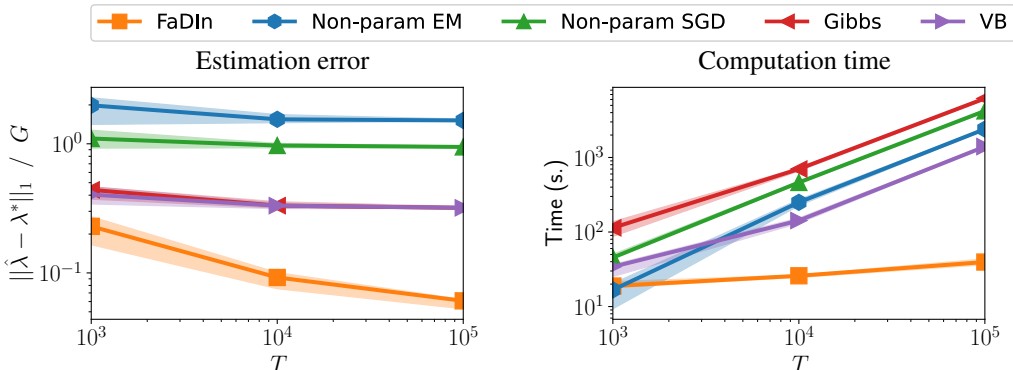

Figure 2: Comparison of the statistical and computational efficiency of FaDIn with four benchmarked methods. The averaged (over ten runs) statistical error on the intensity function (left) and the computational time (right) are computed regarding the time $T$ (and thus the number of events).

## 4    APPLICATION TO MEG DATA

Electrophysiology signals recorded with M/EEG contain recurring prototypical waveforms that can be related to human behavior (Shin et al., 2017). Convolutional Dictionary Learning (CDL; Jas et al. 2017) is an unsupervised method to efficiently extract such patterns and study them in a quantitative way. With CDL, multivariate neural signals are represented by a set of spatio-temporal patterns, called *atoms*, with their respective onsets, called *activations*. Here, we make use of the `alphacsc` software for CDL with rank-1 constraint (Dupré la Tour et al., 2018), as it includes physical priors for the patterns to recover, namely that the spatial propagation of the signal from the brain to sensors is linear and instantaneous. The schema in Figure A.14 in the Appendix presents how CDL applies to MEG recordings.

Experiments on magnetoencephalography (MEG) data were run on two datasets from the MNE Python package (Gramfort et al., 2013; 2014): the *sample* dataset and the somatosensory (*somato*) dataset[1]. These datasets were selected as they elicit two distinct types of event-related neural activations: evoked responses which are time-locked to the onsets of the stimulation, and induced responses which exhibit larger random jitters. The *sample* dataset contains M/EEG recordings of a human subject presented with audio and visual stimuli. This experiment presents checkerboard patterns to the subject in the left and right visual field, interspersed with tones to the left or right ear. The experiment lasts about $4.6 \, \text{min}$, and approximately $70$ stimuli per type are presented to the subject. For the *somato* dataset, a human subject is scanned with MEG during $15 \, \text{min}$, while $111$ stimulations of his left median nerve were made.

For both datasets, raw data are first preprocessed as done by Allain et al. (2021), and CDL is then applied: 40 atoms of duration $1 \, \text{s}$ each are extracted on the *sample* dataset, and 20 atoms of duration $0.53 \, \text{s}$ for the *somato* dataset. Finally, each dataset is represented by two sets of Temporal Point Processes: a set of stochastic ones representing atoms' activations, and a set of deterministic ones coding for external stimuli events. The main goal of applying the TPP framework to such data

---

[1]Both available at https://mne.tools/stable/overview/datasets_index.html

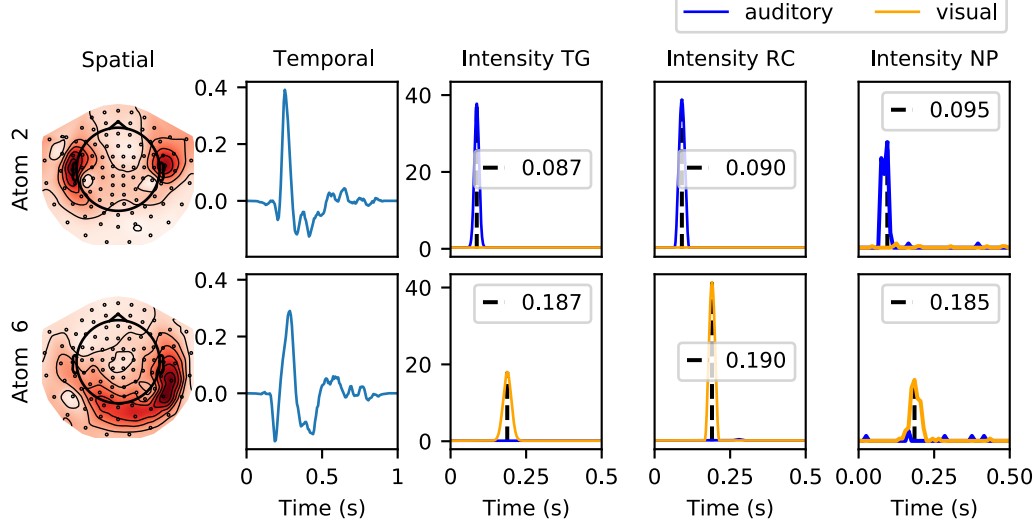

Figure 3: Spatial and temporal patterns of 4 atoms from *sample* dataset, and their respective estimated intensity functions following a stimulus (cue at time = 0 s), for auditory and visual stimuli with non-parametric kernel (NP) and two kernel parametrizations: Truncated Gaussian (TG) and Raised Cosine (RC).

is to characterize directly when and how each stimulus is responsible for the occurrence of neural responses, especially by estimating the distribution of latencies. We are interested in the paradigm of Driven Point Process (DriPP; Allain et al. 2021) and for every extracted atom, its intensity function related to the corresponding stimuli is estimated using a non-parametric kernel (NP) and two kernel parametrizations: Truncated Gaussian (TG) and Raised Cosine (RC). Results on the *sample* (resp. *somato*) dataset are presented in Figure 3 (resp. Figure A.15 in the Appendix), where only the kernel related to each type of stimulus is plotted, for the sake of clarity. See Appendix A.5 for more details.

Results show that all three kernels agree on a peak latency around $90\,\mathrm{ms}$ for the auditory condition and $190\,\mathrm{ms}$ for the visual condition. Due to the limited number of events, one can observe that the non-parametric kernel estimated is less smooth, with spurious peaks later in the interval. Overall, these results on real MEG data demonstrate that our approach with a RC kernel parametrization is able to recover correct latency estimates even with the discretization of stepsize 0.02. Furthermore, the usage of RC allows us to have sharper peaks in the intensity compared to TG, enforcing the link between the external stimulus and the atom's activation. This difference mainly comes from the fact that RC does not need to have its support pre-determined. This advantage is even more pronounced in the case of induced responses, such as in the *somato* dataset (see Figure A.15), where the range of possible latency values is more difficult to determine beforehand.

## 5  DISCUSSION

This work proposed an efficient approach, FaDIn, to infer general parametric kernels for Multivariate Hawkes processes. Our method makes the use of parametric kernels computationally tractable, beyond exponential kernels. The development of FaDIn is based on the three key features: (*i*) finite-support kernels, (*ii*) timeline discretization and (*iii*) precomputations reducing the computational cost of the gradients. These allow for a computationally efficient gradient-based approach, improving state-of-the-art methods while providing flexible use of kernels well-fitted to the considered applications. Moreover, this work shows that the bias induced by the discretization is negligible, both theoretically and numerically. By allowing the use of a general parametric kernel in Hawkes processes, this contribution opens new possibilities for many applications. This is the case with M/EEG data, where estimating information about the rate and latency of occurrences of brain signal patterns is at the core of neuroscience questions. Therefore, FaDIn makes it possible to use a Raised Cosine kernel, allowing for efficient retrieval of these parameters.

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
