# OpenReview forum: "FaDIn: Fast Discretized Inference for Hawkes Processes with General Parametric Kernels"
_ICLR.cc/2023/Conference — Submitted to ICLR 2023_

### Official Review · Reviewer_GU8n · 2022-10-22

**Confidence:** 4
**Correctness:** 3
**Technical Novelty And Significance:** 1
**Empirical Novelty And Significance:** 2
**Recommendation:** 3

**Clarity, Quality, Novelty And Reproducibility:**

Clarity: The paper is easy to follow.

Quality: The derivation of inference seems correct.

Novelty: The submission is lacking in novelty.

Reproducibility: I did not check the code.

**Strength And Weaknesses:**

Strength: The paper is easy to follow and the derivation of inference seems correct but I did not check it carefully.

Weakness: The submission is lacking in novelty and the experiment is not convincing.

**Summary Of The Paper:**

The submission provided a TPP model estimation method based on a L2 loss by assuming parametric finite-support kernel, discretization and precomputation. The statistical and computational efficiency of the method is compared to a nonparametric method. Finally, the method is applied to the MEG data.

**Summary Of The Review:**

The major concern of the submission is novelty. All key ideas of the proposed method: L2 loss, finite-support, discretization and precomputation, are all existing in previous works. There is nothing new in the methodology in this submission. Another conceren is that the experiment is not convincing. The submission compares the statistical and computational efficiency of the proposed method to a nonparametric method, and shows the proposed method is better w.r.t. accuracy and efficiency. On the one hand, the simulated data is from a parametric model and the proposed method also assumes the same parametric form for the kernel, of course the ground-truth parametric model would get a more accurate estimation with limited data; on the other hand, the efficiency of nonparametric methods are certainly less efficient than parametric methods in general. The experiment section does not provide any meaningful comparision.

---

> ### Author Response · Authors · 2022-11-16
> **Answer to Reviewer GU8n**
>
> We thank the reviewer for their comment. While we agree that some effort can be made to provide better experimental results, we believe that our current work is useful for the community. In particular, we provide an efficient and practical solver for discretized MHP, and we study the impact of discretization on estimation bias, which are novel and significant contributions. We are convinced that this rebuttal answers all relevant criticism of our work, and we kindly ask the reviewer to reconsider their rating.
>
>
> **Novelty -** The problem tackled by our method is deemed non-trivial by reviewer EZux and we believe we are the first to provide an analysis --both theoretical and practical-- of the impact of discretization on parameter estimates. While the usage of the discretization and finite kernel supports has been used in previous works, it has not been proposed together with the L2 loss to design a framework adapted to any finite support parametric kernels and to allow further precomputations. Also, we would like to emphasize that this allows FaDIn to scale linearly with the number of events in the precomputation phase and to be independent of the number of events when inferring the kernel parameters. A benchmark comparison has been added to the new version of the paper to highlight this appealing property, see Section 3.2.
>
>
> **Improving experiments -** The new version of the paper has been reworked to include more convincing experiments. In Section 3.2, a statistical and computational comparison with many non-parametric methods (including Bayesian methods), depending on the availability of their implementation, has been added to the new version of the paper and supports this claim. Indeed, it shows that non-parametric benchmarked methods computationally scale linearly with the parameter $T$, which grows roughly as the number of events. In contrast, FaDIn depends on the size of the grid in the precomputation time only and then scales very well with an increasing number of events.
> Following the reviewer’s recommendation, more diversity in the kernel shape has been added in the Section A.2.2 including the Truncated Gaussian and the Truncated Exponential kernels, supporting our claims.
>
>
> - “the simulated data is from a parametric model and the proposed method also assumes the same parametric form for the kernel, of course the ground-truth parametric model would get a more accurate estimation with limited data”
>
>
> **Parametric model -** Indeed, our synthetic experiments are all with a given model. But we need a model to be able to simulate the data and evaluate the model, which is what is usually done in the parametric literature with exponential kernels. But we also provide an application to data with no assumption on its model. Our work is motivated by neuroscience applications, where only limited data are available and require interpretability of the underlying model. This is the purpose of Section 4 to compare the performance of FaDIn on real-world datasets from neuroscience. In these experiments, we do not assume any kind of parametric form for the kernel. As shown in these experiments, FaDIn can characterize directly when and how each stimulus is responsible for the occurrence of neural responses, especially by estimating the distribution of latencies, which is a challenging neuroscience task (see Figures 3 and A.14).

---

### Official Review · Reviewer_h6fy · 2022-10-24

**Confidence:** 4
**Correctness:** 2
**Technical Novelty And Significance:** 3
**Empirical Novelty And Significance:** 2
**Recommendation:** 5

**Clarity, Quality, Novelty And Reproducibility:**

- The text is mostly clear and well-written.
- However, technical clarity and quality is not clear because of the lack of detailed discussions on the theoretical basis and relative comparison with the existing binned Hawkes models.

**Strength And Weaknesses:**

**Strength**

- Addresses the important issue of computational issue of the multivariate Hawkes process.
- Developed a practically feasible procedure.
- Provided a theoretical analysis on the L2 approach.

**Weakness**

- The theoretical basis is not very clear. Especially the justification of the main loss function is not clearly provided. Chapter 2 of Bompaire (2019) does not explicitly derive Eq.(2). The text looks misleading.

- Introduciton misses a major part of the efforts that have been made so far to address the computational issues. Notably,  an EM-like iterative scheme, first proposed by Veen and Schoenberg, has been known as one of the standard approaches to address the issue. But apparently, little attention is paid to that thread.
	> Veen, Alejandro, and Frederic P. Schoenberg. "Estimation of space–time branching process models in seismology using an em–type algorithm." Journal of the American Statistical Association 103.482 (2008): 614-624.



- Also, I don't think it is true that existing multivariate Hawkes processes fail to estimate the decay parameter. In fact, many recent works such as the below seem to estimate the decay parameter without significant issues with the aid of the Veen-Schoenberg algorithm.
	>Idé, T., Kollias, G., Phan, D., & Abe, N. (2021). Cardinality-Regularized Hawkes-Granger Model. Advances in Neural Information Processing Systems, 34, 2682-2694.


- Finally, it is not very clear how the proposed method is compared with the existing binned Hawkes process, in particular Kirchner's INAR(p) model.

**Summary Of The Paper:**

This paper proposes an approximate version of the multivariate Hawkes process. The authors claim that the existing estimation procedure has significant limitations, especially in the computational cost. The authors propose a variant of the binned Hawkes process, where the continuous decay function is replaced by a discrete vector.


**Summary Of The Review:**

The paper is overall well-written. However, the technical novelty is not clearly justified by the authors not only empirically but also methodologically.

---

> ### Author Response · Authors · 2022-11-16
> **Answer to Reviewer h6fy**
>
> We thank the reviewer for their constructive comments and for acknowledging the relevance of our study. While we agree that some efforts can be made to consider existing works better, we believe that the manuscript has been revised to tackle this point. We are convinced that this rebuttal answers all relevant criticism of our work, and we kindly ask the reviewer to reconsider their rating.
>
>
> - “the justification of the main loss function is not clearly provided.”
>
>
> **Choice of loss function -** We thank the reviewer for pointing out this typo, as indeed, the reference associated to Eq. (2) is not in Chapter 2 of Bompaire, 2019, but is Eq. I.2 in Chapter 1, that refers to the L2 norm in the univariate case, but the generalization to multivariate can be done easily. This loss has been justified with risk minimization approaches, for instance in Reynaud-Bouret et al. (2010) or in Hansen et al. (2015), as mentioned at the end of the first paragraph of the introduction. The primary motivation for the L2 loss is the presence of terms that, in contrast to the log-likelihood, can be precomputed (see e.g., Chapter 1 in Bompaire, 2019). In the continuous case, only the precomputation phase depends quadratically on the number of events, which is useful when the number of events is high. When coupled with the discretization as in FaDIn, the L2 loss is further *linear* in the number of events even in the precomputation phase and relies only on the grid parameters. FaDIn leverages finite support kernels with these appealing properties and is then computationally efficient, as shown in our previous experiments and the newly provided benchmark. Empirical comparison with max likelihood-based approaches, as well as Bayesian ones, is now provided in the new benchmark experiments.
>
>
> - “Introduction misses a major part of the efforts that have been made so far to address the computational issues.”
>
>
> **Missing references -** We thank the reviewer for pointing out these references that we had overlooked. The mentioned reference has been added to the manuscript. However, it is worth noting that Veen et al. (2008) is limited to exponential kernels, while FaDIn provides a computationally and statistically efficient framework for the use of *any parametric kernels*. Furthermore, FaDIn is the first approach being independent of the number of events, up to linear precomputations, leading to a computationally efficient approach. In the revised manuscript, a benchmark has been added to better compare with other existing non-parametric approaches. It demonstrates the computational and statistical advantages of our approach, supporting the previous claim.
>
>
> - “I don't think it is true that existing multivariate Hawkes processes fail to estimate the decay parameter.”
>
>
> **Decay parameter estimation -** We thank the reviewer for pointing out Idé et al. (2021), which has been added to the new version of the paper. As pointed out in [1], recovering the MHP parameters by minimizing the Likelihood-Loss (and the L2 loss) when the decay parameter is not fixed is not efficient as the Markov property cannot be used anymore. Therefore, this problem is known to be challenging in the MHP literature. This optimization is even more difficult when the parameter space increases. We agree that our statement should be updated to say that recent approaches have made progress for exponential kernels, such as in [2]. However, to our knowledge, for general kernels such as Raised Cosine or Truncated Gaussian, FaDIn is the first to provide an efficient recovery of the parameters.
>
> - “Finally, it is not very clear how the proposed method is compared with the existing binned Hawkes process, in particular Kirchner's INAR(p) model.”
>
> **Comparison with Kirchner’s INAR(p) -** The approach provided in Kirchner [3] is close to our work. However, there are several key differences between the two algorithms.
>
> First, FaDIn directly optimizes the L2 loss of the Hawkes process model, while Kirchner method resort to indirect computation through solving an INAR(p) model and then re-estimating the kernel with an ad hoc procedure, potentially leading to sub-optimal estimation. Moreover, the results by Kirchner only show convergence of the INAR(p) model to Hawkes processes, while we quantify the convergence speed of FaDIn’ parameters in Proposition 2. In contrast to FaDIn, Kirchner's approach allows kernel estimation on a specific grid only, as pointed out in https://arxiv.org/pdf/1509.02017.pdf page 15, leading to a more general estimation procedure.
>
> References.
>
> [1] R. Lemonnier and N. Vayatis. Nonparametric Markovian Learning of Triggering Kernels for Mutually Exciting and Mutually Inhibiting Multivariate Hawkes Processes (2014).
>
> [2] Idé, T., Kollias, G., Phan, D., & Abe, N. Cardinality-Regularized Hawkes-Granger Model. Advances in Neural Information Processing Systems, 34, 2682-2694 (2021).
>
> [3] M. Kirchner. An estimation procedure for the Hawkes process (2017).

---

### Official Review · Reviewer_mV6v · 2022-10-25

**Confidence:** 3
**Clarity, Quality, Novelty And Reproducibility:** See above.
**Correctness:** 3
**Technical Novelty And Significance:** 2
**Empirical Novelty And Significance:** 2
**Recommendation:** 5

**Strength And Weaknesses:**

**Strengths**
- Theoretical analysis of discretization impact may be of interest to some researchers
- The proposed approach improves computational efficiency
- The proposed approach seems easy to implement

**Weaknesses**

*Technical limitations*
- The proposed approach is limited to parametric self-exciting Hawkes kernels which are often violated in practice
- The proposed approach could be sensitive to the selected discretization step-sizes $\Delta$ and the length of kernel's support $W$
- The novelty is quite limited. Discretizing and limiting support of existing parametric kernels seems straightforward

*Lack of clarity*
- In general, the writing and notation should be improved to increase clarity, e.g., it's not necessary to write kernel convolution with Dirac functions since it's the kernel evaluated at that location.

*Experimental Limitations*
- Figure 1: For easy comparisons, the paper should plot all methods on the same plot. As is,  the y-axis is on a different scale making it hard to make comparisons.
- The paper should consider adding additional real-world datasets and recent state-of-the-art  baselines (*e.g.*, neural Hawkes [1] models flexible kernels with neural networks)
- Figure 3:  Are the TG and RC intensity predictions from the proposed approach?
- The performance improvements over baselines seem marginal
- The paper should provide summary statistics of the synthetic and electrophysiology, *e.g.*, number of event types, sequence lengths, number of sequences, *etc.*
- The paper should provide sensitivity analysis for step-sizes $\Delta$ and length of kernel's support $W$

**References**
- [1] Mei et al., "The neural hawkes process: A neurally self-modulating multivariate point process", NeurIPS, 2017

**Summary Of The Paper:**

The paper proposes discretizing and limiting kernel support for efficiently learning kernel parameters of a Hawkes process via gradient descent. Additionally, the paper provides theoretical insights measuring the discretization bias. Experimental results on synthetic and electrophysiology datasets show improved estimation of kernel parameters and computation time compared to baselines.

**Summary Of The Review:**

The technical novelty is limited and experimental analysis is unconvincing.

---

> ### Author Response · Authors · 2022-11-16
> **Answer to Reviewer mV6v**
>
> We thank the reviewer for their detailed review and for judging our paper interesting. We agree that some efforts can be made to better compare our method to existing work. However, we are convinced that the present rebuttal is answering all relevant criticism to our work, which hopefully will convince the reviewer to further increase their rating.
>
>
> - “The novelty is quite limited. Discretizing and limiting support of existing parametric kernels seems straightforward.”
>
>
> **Novelty -** The problem tackled by our method is deemed interesting and non-trivial by reviewers EZux and h6fy, and we believe we are the first to provide an analysis --both theoretical and practical-- of the impact of discretization on parameter estimates. While the usage of the discretization and finite kernel supports has been used in previous works, it has not been proposed together with the L2 loss to design a framework adapted to any finite support parametric kernels and to allow further precomputations. Also, we would like to emphasize that this allows FaDIn to scale linearly with the number of events in the precomputation phase and to be independent of the number of events when inferring the kernel parameters. A benchmark comparison has been added to the new version of the paper to highlight this appealing property, see Section 3.2.
>
>
> - “The proposed approach could be sensitive to the selected discretization step-sizes Δ  and the length of kernel's support W”
> - “The paper should provide sensitivity analysis for step-sizes Δ and length of kernel's support W”
>
>
> **Sensitivity to $\Delta$ and $W$ -** Indeed, the discretization parameter $\Delta$ and kernel’s length $W$ are two important hyperparameters of our method.
> Regarding sensitivity analysis for $\Delta$, we think that Figures 1 and A.6–12 provide good insights of its influence onto the estimation error for multiple kernel types. However, sensitivity analysis for $W$ is indeed missing. We thus performed such a study, and it is now presented in Appendix A.1. As explained in this part, the statistical accuracy of the parameters estimated with finite support kernels of length $W$ converges quickly to those of the infinite size kernel (see Appendix A.1 for more details).
>
>
> - “the writing and notation should be improved to increase clarity”
>
>
> **Writing clarity -** Efforts have been made in the new version of the manuscript to improve clarity of the paper. If some further points need clarification, we kindly ask the reviewer to point us toward them.
>
> - “Figure 1: For easy comparisons, the paper should plot all methods on the same plot.”
>
>
> **Comment on Figure 1 -** We thank the reviewer for their suggestion. For clarity purposes, we have kept the current figures in the main manuscript as we believe they are easier to read, but we have added a figure, see Figure A.5, to show comparison between the two. It highlights the consistency advantages of FaDIn over the EM algorithm.
>
>
> - “The paper should consider adding additional real-world datasets and recent state-of-the-art baselines”
>
>
> **Comparison with more baselines -** We have added a benchmark comparing three other existing algorithms with open source implementations as baselines in Figure 2 of the revised manuscript. As for real world dataset experiments, the presented experiments aim to expand the application range of PP to other contexts with longer range interaction and we believe the presented application illustrates this point well. Our experiments on neural data aim to show we are able to capture specific effects -- i.e. delays in the brain -- with a low number of events, which is currently not possible with existing methods. As our goal is not to provide a new model that best fits other real-world datasets, we believe that extra comparisons, showing the value of the loss on hold out data, would not really be informative for the reader. But if the reviewer thinks we should add a specific dataset, we would be happy to add it in our comparisons.
>
> - "Figure 3: Are the TG and RC intensity predictions from the proposed approach?"
>
> **TC and RC intensity in figure 3 -** The intensities for RC are indeed computed using the proposed approach, whereas, for the TG ones, they have been calculated using continuous EM, as done by Allain et al. (2021) and serve as a baseline comparison. The advantage of using FaDIn is using a Raised Cosine kernel with discretization and self-learning support. Indeed,  the previous EM algorithm was unable to handle such a kernel. Further, we obtain similar results in terms of latency values (which is expected) while being less prone to variance.
>
>
> - "The paper should provide summary statistics of the synthetic and electrophysiology"
>
> **Summary statistics for the considered datasets -** We thank the reviewer for this suggestion, we now provide summary statistics of the datasets used in the Appendix (see Table A.1).

---

### Official Review · Reviewer_rbja · 2022-10-28

**Confidence:** 3
**Correctness:** 3
**Technical Novelty And Significance:** 2
**Empirical Novelty And Significance:** 2
**Recommendation:** 5

**Clarity, Quality, Novelty And Reproducibility:**

Overall, the paper is written well and easy to follow. The experimental section may be improved with more baselines and better clarity in the  claims.
In terms of the novelty, similar models have been proposed in the past (as mentioned above), it would be great to compare and contrast with them too.

**Strength And Weaknesses:**

Strengths
- the paper is well written and easy to follow.
- The impact of discretization is studied well, theoretically as well as experimentally

Weakness / Questions / Comments
- Discrete time Hawkes processes are also considered in some of the earlier works such as https://www.cs.princeton.edu/~rpa/pubs/linderman2015scalable.pdf, https://arxiv.org/pdf/2003.02810.pdf
It would be better to compare the methodologies in this submission agianst them, theoretically as well as in experiments.

- The experiments section may be improved with more convincing claims. For instance, in kernel recovery comparisons, only a single kernel is used for generating the samples. For fair comparisons, more kernel types should be compared, assuming that FasDin is not aware of the true kernel.

- In Figure 1, it is not clear how the recovery error compares across the models for all choices of T.

- Section 4 can benefit from better description of the data and the setup. What are the choices of \Delta tried? What Is the typical value of T  or G for the samples ? While the latency estimates are similar, what is the specific computational advantage in terms of the total time ?

**Summary Of The Paper:**

In this paper, the authors study discrete Hawkes process to model temporal data, along with parametric kernels with fixed support.
They propose to use least squares loss to learn the model parameters (compared to the more popular log likelihood estimation). They propose to use gradient descent, where the gradients may be efficiently computed thanks to the discreteness of the process and the finite support of kernels. They also derive the results for the approximated solution to converge to the true solution as the discretization window approaches zero. Numerical results are presented on (a) simulated data to evaluate the kernel learning and to estimate the recovery error as a function of \Delta (the bin width)  (b) MEG data illustrating the learnt latency.

**Summary Of The Review:**

Overall, the paper has good technical contributions, but may be improved significantly with convincing experimental claims.

---

> ### Author Response · Authors · 2022-11-16
> **Answer to reviewer rbja**
>
> We thank the reviewer for their constructive comments and for acknowledging our efforts to validate the approach. While we agree that some efforts can be made to better compare our method to existing work, we believe that our current work improves on the state-of-art and also is useful for the community. Thus, we kindly ask the reviewer to reconsider their rating as we believe the required clarification has been included in the revised manuscript.
>
>
> - “Discrete time Hawkes processes are also considered in some of the earlier works”
>
>
> **Discrete time Hawkes processes -** We thank the reviewer for pointing out such interesting papers that we had unfortunately missed. Bayesian literature, including this paper, has been added to the introduction in the new version of the paper. Empirically, we compared the two approaches derived in this paper and two additional non-parametric methods to FaDIn. This benchmark, highlighting the statistical and computational advantages of FaDIn over these approaches, is now depicted in Figure 2 of the revised paper.  Indeed, it shows a better recovery of the parameters of the underlying true model with a lower computation time, for different sizes of T (and then the number of events). It is worth noting that we provide the first approach being *linearly dependent* on the number of events for any parametric kernels.
> From a theoretical perspective, the aforementioned paper does not give any results on their method. In contrast, we provide theoretical convergence of FaDIn towards the continuous version of FaDIn in Section 3.1.
>
>
> - “The experiments section may be improved with more convincing claims.”
>
>
> **Improving experiments -** The new version of the paper has been reworked to include more convincing experiments. In Section 3.2, a statistical and computational comparison with many non-parametric methods (including Bayesian methods), and for three kernel shapes (see Section A.2.1), , depending on the availability of their implementation, has been added to the new version of the paper. These results support our claim. Indeed, it shows that non-parametric benchmarked methods computationally scale linearly with the parameter T, which grows roughly as the number of events. In contrast, FaDIn depends on the size of the grid in the precomputation time only and then scales very well with an increasing number of events.
> Following the reviewer’s recommendation, Additional kernel shapes have been added in the Section A.2.2 including Truncated Gaussian and Truncated Exponential, supporting our claims.
>
>
> - “In Figure 1, it is not clear how the recovery error compares across the models for all choices of T.”
>
>
> **Figure 1 clarification -** We are not totally sure of what the reviewer means here. We think that the reviewer is asking about a comparison between both sides of the figures.  For clarity purpose, we have kept the current one as we deem it more readable but we have added Figure A.5, to show comparison between the two. It highlights the consistency advantages of FaDIn over the EM algorithm.
>
>
> - “Section 4 can benefit from better description of the data and the setup.”
>
>
> **Data description in section 4 -** Thanks for pointing out this overlook. Due to space limitations, we have kept the description of the datasets brief. Still, we have added a section in the Appendix to add more information. Following the recommendation of the mV6v reviewer, we now provide summary statistics of the datasets used in Appendix (see Table A.1).
>
>
> - “What are the choices of \Delta tried? What Is the typical value of T or G for the samples ?”
>
>
> **Choices of $\Delta$ -** Regarding Figure 3 on real MEG datasets, the data are already on a natural grid as they are acquired with a given sampling rate of 150 Hz. We chose $\Delta$ accordingly to this rate to match the grid of the CDL. For this dataset, the duration of the recording $T$ is 4.6 minutes. For the “somato” dataset presented in Figure A.14, we kept the same value of $\Delta$, and the duration of the MEG recording was 15 minutes. Please note that those two standard datasets contain the recording of one subject each. This complementary information has been clarified in the new version of the paper.

---

> > ### Author Response · Authors · 2022-11-16
> > **Additional answer**
> >
> > - "While the latency estimates are similar, what is the specific computational advantage in terms of the total time ?"
> >
> >
> > **Total time for real MEG data -** On the presented applications, the total time of FaDin is 3.7 s. compared to EM with truncated Gaussian kernel which is 21.8 s on the sample datasets for instance. The real advantage of our method is that we can use general kernel parameterizations that the previous EM algorithm was unable to handle. It allows to obtain similar results in terms of latency values (which is expected) while being less prone to variance. Moreover, for further real MEG applications such as epilepsy data, where recordings range in hours, the presented method will be able to scale and provide reliable results efficiently. See https://doi.org/10.1371/journal.pone.0275063 as an example of work extracting spike events on clinical MEG.

---

### Official Review · Reviewer_EZux · 2022-11-02

**Confidence:** 5
**Correctness:** 2
**Technical Novelty And Significance:** 3
**Empirical Novelty And Significance:** 2
**Recommendation:** 3

**Clarity, Quality, Novelty And Reproducibility:**

The work is in general well presented. However there are some typos and the writing is not always clear in terms of explanations or  how the notation is introduced. The assumptions and relaxations the authors make are reasonable but not well justified in all cases and statements which support these choice are not always correct (see above). This compromises the correctness of the work.
There is enough information on how the experiments were made and supplementary material.

**Strength And Weaknesses:**

Strengths of the paper are firstly the efficient computation for multivariate hawkes processes which is a non-trivial task. To achieve this the authors make some assumptions and relaxations such as assuming aggregated rather than exact events and finite support. They choose to minimize a least square loss which they achieve efficiently with some pre-computations. The authors motivate well their proposal and the benefit of hawkes process efficient inference is clear especially withing the neuroscience context. The application is both relevant and interesting.

The paper disregards bayesian inference approaches to the problem - and there are quite a lot both in previous and in the recent literature covering both exact and aggregated approaches. Many authors have proposed different approaches to multivariate hawkes processes that in fact rely on less approximations than those proposed here. For instance when it comes to efficient computation of discretized hawkes processes approaches already exist (G. Mohler 2013).
Additionally when it comes to non-parametric kernel approaches, there are way more than the paper suggests (see papers by J. Rousseau). In the context of non-parametric vs parametric approaches, the authors claim throughout that parametric approaches are less prone to over-fitting in comparison to parametric ones and use this as an advantage of their parametric kernels. However this is not true. If the authors wanted to make a statements about the potential advantages of parametric approaches vs nonparametric ones, these could be around other topics such as model mismatch. Other claims that are not true are for instance that Hawkes processes are easy to fit among other temporal point processes. The contrary, considering homogeneous or other inhomogeneous poisson processes often used to explain temporal or spatial patterns, hawkes processes are not striaghtforward to fit.
The choice of the loss function is not well motivated nor justified and should be compared theoretically and empirically with max likelihood-based approaches, as well as bayesian ones ideally. The loss function choice should not be arbitrary (or merely serving computational purposes) but should be properly justified in regards to the model being used.



**Summary Of The Paper:**

The authors deal with the problem of performin inference for multivariate hawkes processes with parametric kernels. They propose an efficient discretized inference scheme that aims to optimize the least square objective (rather than max likelihood for instance) which does not assume exact events but aggregated counts, hence the name discretized inference. Their approach covers parametric kernels and the efficiency of the compution relies on pre-computations and on finite supports. Finite supports means that they discard events in the long past as they claim these have low infleunce for the scope of their applications being to quantify the dependence between neural response and external stimuli.

**Summary Of The Review:**

Overall this paper targets a known problem, that of computationally expensive inference for multivariate hawkes processes. The authors motivate their work within the context of neuroscience in which hawkes process can find great use and can bring interesting conclusions on neural activity. The assumptions the aithors make to achieve their computationally efficient inference are reasonable but the choice of loss function is not well motivated or justified and needs further support (theoretical guarantees, empirical examples on higher dimensions and comparisons). Additionally several claims supporting the proposal choices are incorrect. Overall although the authors make interesting suggestions for a known problem, the paper as it is does not have enough methodological support for a publication.

---

> ### Author Response · Authors · 2022-11-16
> **Answer to Reviewer EZux**
>
> We thank the reviewer for their meticulous feedback and for acknowledging the interest of providing efficient computation for parametric MHP, which is a non-trivial task. While we agree that some of our assumptions could be better discussed and challenged, we believe that our current work is useful for the community. In particular, we provide an efficient and practical solver for discretized MHP, and we study the impact of discretization on estimation bias, which are novel and significant contributions. We are convinced that the present rebuttal is answering their relevant criticisms to our work, which hopefully will convince the reviewer to further increase their rating.
>
> - “The paper disregards Bayesian inference approaches to the problem”
>
>
> **Comparison to Bayesian inference -**  We agree with this comment. A paragraph on Bayesian inference approaches has been added in the introduction of the paper. In contrast to these approaches, FaDIn does not depend on the number of events, thanks to the L2 loss and the pre-computations it provides. A statistical and computational comparison with many non-parametric methods (including Bayesian methods), depending on the availability of their implementation, has been added to the new version of the paper and supports this claim. Indeed, it shows that non-parametric benchmarked methods computationally scale linearly with the parameter T, which grows roughly as the number of events. In contrast, FaDIn depends on the size of the grid in the precomputation time only (and not on the number of events) and then scales very well with increasing events.
>
>
> - “the authors claim throughout that parametric approaches are less prone to over-fitting in comparison to non-parametric ones. However, this is not true.”
>
>
> **Non-parametric approach prone to overfit -** As shown in Figure 2 of the original paper (now moved to Figure A.3), the non-parametric approaches result in noisy estimates of the kernel, with probability mass where the kernel is zero. We agree that the employed term ''over-fitting'' may not be adapted. We removed their occurrence in the new version of the paper and replaced them with ‘’noisy estimates’’.
>
>
> - “Other claims that are not true are for instance that Hawkes processes are easy to fit among other temporal point processes. In the contrary, considering homogeneous or other inhomogeneous Poisson processes often used to explain temporal or spatial patterns, hawkes processes are not straightforward to fit.”
>
>
> **Hawkes processes are easy to fit -** We thank the reviewer for pointing out this inaccuracy. We meant that it was doable with exponential or non-parametric kernels, as opposed to other self-exciting PP. Our wording has been modified in the abstract of the new version of the paper to state that Hawkes processes are widely used due to their adequate modeling properties for various applications.
>
>
> - “The choice of the loss function is not well motivated nor justified and should be compared theoretically and empirically with max likelihood-based approaches, as well as Bayesian ones, ideally.”
>
>
> **Choice of loss function -** The primary motivation for the L2 loss is the presence of terms that, in contrast to the log-likelihood, can be precomputed (see e.g., Chapter 1 in Bompaire, 2019). In the continuous case, only the precomputation phase depends quadratically on the number of events, which is useful when the number of events is high. When coupled with the discretization as in FaDIn, the L2 loss is further *linear* in the number of events even in the precomputation phase and relies only on the grid parameters. FaDIn leverages finite support kernels with these appealing properties and is then computationally efficient, as shown in our previous experiments and the newly provided benchmark. Empirical comparison with max likelihood-based approaches, as well as Bayesian ones, is now provided in the new benchmark experiments.

---

> ### Comment · Reviewer_EZux · 2022-12-01
> **Thanks for the response**
>
> I thank the authors for their response. I can acknowledge that I have read their response on my raised points.

---

### Decision · Program_Chairs · 2023-01-20

**Decision:**

Reject

**Justification For Why Not Higher Score:**

The reviewers were in agreement.

**Justification For Why Not Lower Score:**

N/A

**Metareview: Summary, Strengths And Weaknesses:**

The manuscript tackles an important problem: multivariate temporal point process inference using parametric kernels with finite support to enable more appropriate modelling of various domains, exemplified by neuroscience applications. The proposed solution requires discretizing the data (the events) and makes an important contribution by investigating the effect of discretization on parameter estimates. There were a number of places the reviewers agreed that the manuscript could be improved significantly, and the authors have made a start, by, for example, adding text about Bayesian inference and more experimental evaluation. The manuscript definitely has potential and I strongly encourage the authors to revise it for a future submission.